# Comparative Clustering (CompaCt) of eukaryote complexomes identifies novel interactions and sheds light on protein complex evolution

Joeri van Strien[1], Felix Evers[2], Madhurya Lutikurti[3], Stijn L. Berendsen[3], Alejandro Garanto[3,4,5], Geert-Jan van Gemert[2], Alfredo Cabrera-Orefice[1], Richard J. Rodenburg[5,6], Ulrich Brandt[3,5,7], Taco W. A. Kooij[2], Martijn A. Huynen[1] *

1 Department of Medical BioSciences, Radboud University Medical Center, Nijmegen, the Netherlands, 2 Medical Microbiology, Radboud Center for Infectious Diseases, Radboud University Medical Center, Nijmegen, The Netherlands, 3 Department of Pediatrics, Amalia Children's Hospital, Radboud University Medical Center, Nijmegen, the Netherlands, 4 Department of Human Genetics, Radboud University Medical Center, Nijmegen, The Netherlands, 5 Radboud Center for Mitochondrial Medicine (RCMM), Radboud University Medical Center, Nijmegen, the Netherlands, 6 Department of Pediatrics, Translational Metabolic Laboratory, Radboud University Medical Center, Nijmegen, the Netherlands, 7 Cologne Excellence Cluster on Cellular Stress Responses in Aging-Associated Diseases (CECAD), University of Cologne, Cologne, Germany

* Martijn.Huijnen@radboudumc.nl

**Data Availability Statement:** All complexome profiling datasets generated and used in this study are available on the CEDAR database (https://

## Abstract

Complexome profiling allows large-scale, untargeted, and comprehensive characterization of protein complexes in a biological sample using a combined approach of separating intact protein complexes e.g., by native gel electrophoresis, followed by mass spectrometric analysis of the proteins in the resulting fractions. Over the last decade, its application has resulted in a large collection of complexome profiling datasets. While computational methods have been developed for the analysis of individual datasets, methods for large-scale comparative analysis of complexomes from multiple species are lacking. Here, we present Comparative Clustering (CompaCt), that performs fully automated integrative analysis of complexome profiling data from multiple species, enabling systematic characterization and comparison of complexomes. CompaCt implements a novel method for leveraging orthology in comparative analysis to allow systematic identification of conserved as well as taxon-specific elements of the analyzed complexomes. We applied this method to a collection of 53 complexome profiles spanning the major branches of the eukaryotes. We demonstrate the ability of CompaCt to robustly identify the composition of protein complexes, and show that integrated analysis of multiple datasets improves characterization of complexes from specific complexome profiles when compared to separate analyses. We identified novel candidate interactors and complexes in a number of species from previously analyzed datasets, like the emp24, the V-ATPase and mitochondrial ATP synthase complexes. Lastly, we demonstrate the utility of CompaCt for the automated large-scale characterization of the complexome of the mosquito *Anopheles stephensi* shedding light on the evolution of metazoan protein complexes. CompaCt is available from https://github.com/cmbi/compact-bio.

www3.cmbi.umcn.nl/cedar/). The source code of the complete comparative clustering software is available on github https://github.com/cmbi/compact-bio).

**Funding:** JS and ACO were supported by the Netherlands Organization for Health Research and Development (ZonMW; TOP 91217009), awarded to MAH and UB. ML was supported by a TOP Grant from the Netherlands Organization for Scientific Research (NWO; TOP 714.017.00 4), awarded to UB. FE and TWAK were supported by the Netherlands Organisation for Scientific Research (NWO-VIDI 864.13.009), awarded to TWAK. The funders had no role in study design, data collection and analysis, decision to publish, or preparation of the manuscript.

**Competing interests:** The authors have declared that no competing interests exist.

## Author summary

Proteins carry out essential functions in the majority of processes in life, often by binding with other proteins to form multiprotein complexes. State of the art experimental techniques such as complexome profiling enable large-scale identification of protein complexes in a biological sample. With the increase in the use of this method in recent years these experiments have been performed on a variety of species, the results of which are publicly available. Combining the results from these experiments presents a computational challenge, but could identify novel protein complexes and provide insights into their evolution. Here, we introduce CompaCt as a method to integrate complexome profiles from multiple species enabling automatic large-scale characterization of protein complexes. It identifies commonalities as well as the differences between species. By applying CompaCt to a collection of complexome profiles, we identified candidate complexes and interacting proteins in a number of species that were not detected in previous separate analyses of these datasets. In doing so we shed light on the evolutionary origin of several protein complex members, pinpointed the function of biomedically relevant proteins whose role was previously unknown, and performed the first investigation of the *Anopheles stephensi* complexome, a mosquito that transmits the malaria parasite.

## Introduction

Most biological processes are mediated by proteins, which in many cases need to form multiprotein complexes to carry out their function. The complexome is the complete set of multiprotein complexes present in a biological system, be it an entire organism or a more limited subsystem such as a specific tissue, cell type, organelle or life stage. Complexome profiling, an 'omics approach aimed at large-scale identification of protein complexes in a single experiment [1] has in the last decade enabled untargeted and systematic analysis of complexomes, reviewed in [2]. In a complexome profiling experiment, native protein complexes are systematically separated across a number of fractions that are then separately analyzed by tandem mass spectrometry to assess their content in quantitative manner. Proteins contained in the same complex or subassembly typically co-migrate and exhibit similar abundance profiles across all or a subset of the fractions. This approach allows interrogation of the composition of a large number of protein complexes as well as of their assembly in a given biological sample in an untargeted manner.

In recent years several computational approaches have been developed to infer protein complexes from these data. Some of these rely solely on the complexome profiling data [3–5], while others use a reference of known protein complexes [6–8] or integrate additional types of interaction evidence to improve protein complex identification [7,9,10]. Additionally, several tools allow comparison of multiple complexome profiles from one species to examine the effect of certain mutations or conditions on the composition or assembly of the investigated complexome [8,11,12].

Since the initial application of the complexome profiling approach, an increasing number of complexome profiling datasets have been generated from at least 21 different species, covering all kingdoms of life [2], which are available from the CEDAR database of complexome profiles [13]. Leveraging these data in comparative analyses in principle has the potential for more sensitive detection of novel interactors and to facilitate analyses of the evolution of protein complexes. Nevertheless, comparative analysis of multiple complexome profiles poses a methodological challenge. Firstly, available datasets have been generated by different laboratories,

using different experimental protocols and resolutions for the separation of protein complexes. This renders direct comparison of the "raw" protein migration profile data difficult. Secondly, while integrating complexome profiles from related species has helped to identify evolutionarily conserved protein interactions consistently present in multiple species [9,10], these approaches preclude characterization of species-specific interactors. Large-scale identification of not just conserved, but also taxon-specific complex members and interactions would result in a more complete picture of the investigated complexomes, and would facilitate evolutionary analyses of protein complexes.

To comparatively analyze the protein complexes in multiple complexome profiling datasets, we developed Comparative Clustering (CompaCt), a computational approach that performs fully automated large-scale integration of protein interaction data from multiple species. Our method of utilizing orthology relations allows combined analysis of multiple species in a manner that enables identification of conserved as well as taxon-specific protein complexes and interactions in a single, uniform analysis platform. Additionally, by including multiple interaction datasets per analyzed species CompaCt is able to distinguish consistently co-migrating proteins representing true interactors from spuriously co-migrating proteins that do not interact. Therewith, it improves the reliability of the analysis of complexes from the individual species included in the comparative analysis.

We have applied our tool to a collection of 53 complexome profiling datasets from nine different species covering the major eukaryotic branches. We demonstrate that integrative analysis of these data with CompaCt is able to systematically recover known conserved and taxon-specific interactions and pinpoint novel interactors and complexes missed in previous analyses. Moreover, by presenting the first analysis of the complexome of the malaria transducing mosquito *A. stephensi*, we demonstrate the usefulness of CompaCt for performing large scale and automatic identification of protein complexes to increase our understanding of the evolution of eukaryotic protein complexes.

## Results

### CompaCt: Comparative clustering of interaction datasets

Our comparative clustering (CompaCt) approach performs automated comparative analysis of protein interaction data representing a number of complexomes (i.e., the set of complexes present in a species, cell type, etc.), and enables systematic identification of not just conserved, but also of taxon-specific interactions. To this end, rather than asking whether interactions between orthologous proteins are conserved, which precludes analysis of proteins without known orthologs, we ask whether any two proteins from different species interact with the same set of orthologous proteins. Thus, even though proteomes of different species will only partly overlap, we are still able to estimate whether two proteins are part of the same complex by identifying orthologs among their interactors. An overview of the CompaCt workflow is shown in Fig 1.

CompaCt requires as input datasets with interaction scores between all protein pairs. Any numeric values representing interaction strength or likelihood that allows ranking each protein's interactors can be used (e.g., correlation, machine learning-based scores, etc.). In the specific application of CompaCt used in this project we have used Pearson correlations between the protein migration patterns resulting from complexome profiling as interaction score. To estimate whether two proteins have common interactors, we compare their sets of interactions that are ranked based on their interaction scores. We refer to these ranked sets as their interactor profile. To be able to quantify the similarity between interactor profiles, we use the rank biased overlap (RBO) metric [14]. This metric determines the similarity between

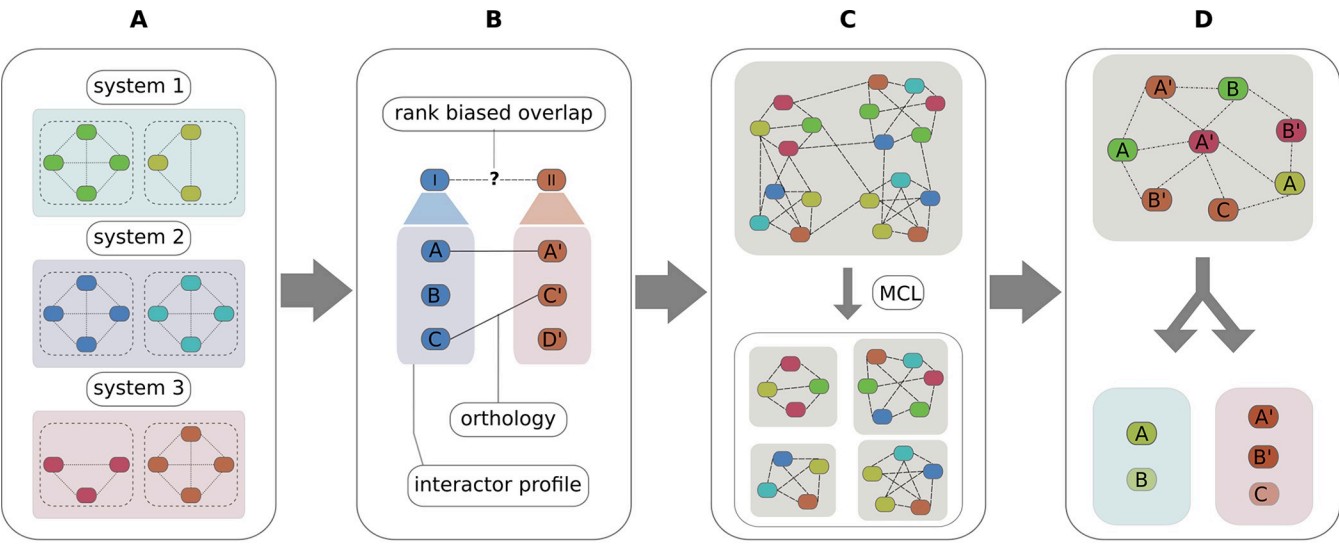

**Fig 1. Overview of CompaCt analysis steps.** (A) Input correlation datasets from multiple biological systems, with lines indicating correlations between proteins based on the similarity in their migration profiles in the complexome profiling data and colors indicating proteins from different datasets. (B) Determining interactor profile similarity of two proteins, I and II, from different datasets, by calculating the overlap between their interactors (A, B, C, etc.), using orthology (A-A', C-C') when they are from different species, with the rank biased overlap metric [14]. (C) Clustering the combined network of all proteins (nodes) with RBO scores (edges) into clusters with MCL [15]. (D) Processing of MCL clusters, separating into subclusters per system (e.g., from the same tissue or species), while pooling information of datasets representing the same system.

ranked lists, while allowing lists of different lengths with non-common elements. Applying this metric to compare interactor profiles we can determine whether any two proteins from different proteomes have common, orthologous interactors, independent of whether they themselves are orthologous or not. To systematically compare interaction datasets, the interactor profile similarity is then computed between all possible protein pairs from all included datasets. The complete set of RBO similarity scores computed between protein pairs form the edges of a large hypernetwork that connects proteins identified across all included datasets. Clusters of connected proteins are extracted from this network using the Markov Cluster (MCL) algorithm [15]. We call the clusters resulting from the MCL analysis superclusters. They contain a mix of proteins from multiple datasets that can originate from different complexomes, e.g. if they come from multiple species, or from multiple subsystems such as tissues, cell types or life stages. The clustered proteins are then sorted into subclusters, each corresponding to one of the included complexomes. To robustly identify interactors, multiple interaction datasets can be included for each complexome. Exploiting the information in these datasets for the same complexome, the clustered proteins are scored by determining the fraction of datasets for which they are part of this cluster. This "fraction clustered" (FrC) score enables prioritization of likely complex members. To prioritize clusters with high probability to present actual protein complexes, the resulting superclusters are scored and filtered based on the similarity between the sets of clustered proteins from different datasets, as determined for all datasets represented in the cluster. The CompaCt software is available as a user-friendly command line tool, as well as a flexible Python package, along with detailed documentation and instructions, from github (https://github.com/cmbi/compact-bio), the python package index (https://pypi.org/project/compact-bio) and dockerhub (https://hub.docker.com/r/joerivanstrien/compact-bio).

## Combined analysis of eukaryote complexome profiles with CompaCt improves recovery of known protein complexes

To comparatively explore the complexomes of eukaryotes, 53 complexome profiling datasets were analyzed with CompaCt. These datasets represent a set of 12 complexomes from various biological systems (e.g.: tissues, life stages, cell-types) in nine different eukaryotic species. A detailed overview of the analyzed complexome profiling data is shown in Table 1. Sensitive pairwise orthology predictions [16] were computed by including best bidirectional hits at the sequence profile level. An overview of the resulting superclusters is presented in Fig 2, showing the size and cluster coherence score of all clusters, including those that did not pass CompaCt's filtering step. The cluster coherence score reflects the degree of commonality between the clustered proteins originating from different datasets. A total of 726 superclusters were identified, each consisting of one or more subclusters that represent one of the included complexomes. 332 of these passed CompaCt's filtering for coherence across datasets. Out of these 332 super-clusters, 254 were consistently represented in more than one complexome. To automatically annotate the results, CompaCt identifies clusters that overlap with a provided set of reference complexes: 36 of the 81 consistently represented human subclusters overlapped with complexes listed in the CORUM database [17] (i.e., contain more than half of the total number of subunits comprising the reference complex). An overview of all clusters that passed filtering, with detailed information, scores, their composition and automatic annotation is available in S1 Data.

CompaCt identifies complexome-specific clusters after first integrating data from multiple complexomes into a single network. To determine whether the combined analysis of multiple species with CompaCt improves recovery of protein complexes for a single species when compared to independent, species-specific analyses, we computed the agreement of human sub-clusters with CORUM [17] while progressively including additional complexomes in the analysis (Fig 3). To compute the agreement of the human subclusters resulting from CompaCt with the CORUM reference, we use the maximum matching ratio metric (MMR), a score that computes overlap with a reference based on a mapping between the resulting clusters and the reference complexes [27]. Starting from separate analysis of just the datasets from the human

**Table 1. Overview of analyzed complexomes.**

| Alias | Species | Sample | n | Proteins | Fractions | CEDAR study id(s) | Reference(s) |
|---|---|---|---|---|---|---|---|
| HUM | *H. sapiens* | Fibroblast | 8 | 5630 | 60 | CRX22 CRX17 CRX15 CRX9 CRX8 | [18–22] |
| BOVIN | *B. taurus* | Heart | 6 | 1180 | 60 | CRX33 | [23] |
| ANOST | *A. stephensi* | Salivary tissue | 2 | 925 | 60 | CRX41 | This paper |
| YARLI | *Y lipolytica* | | 4 | 3515 | 60 | CRX40 | - |
| PF3_GAM | *P. falciparum* | Gametocyte | 4 | 1672 | 60–61 | CRX23 | [24] |
| PF3_AS | *P. falciparum* | Asexual stage | 4 | 1732 | 60–61 | CRX23 | [24] |
| PF3_SCH | *P. falciparum* | Blood-stage schizont | 6 | 1816 | 48 | CRX20 | [9] |
| BER_SCH | *P. berghei* | Blood-stage schizont | 6 | 1553 | 48 | CRX20 | [9] |
| KNO_SCH | *P. knowlesi* | Blood-stage schizont | 6 | 1885 | 48 | CRX20 | [9] |
| TOX | *T. gondii* | Tachyzoite | 1 | 839 | 61 | CRX27 | [25] |
| AT_LF | *A. thaliana* | Leaf | 3 | 1518 | 30 | CRX24 | [26] |
| AT_SD | *A. thaliana* | Seedling | 3 | 1957 | 30 | CRX24 | [26] |

Overview of analyzed complexomes, with descriptions of the datasets included per complexome. Note that a number of analyzed complexomes were specifically designed to focus on the mitochondrial component. Despite this, most of these datasets also include proteins and complexes from other cellular compartments. For more detailed descriptions of the specific datasets, please refer to the methods, respective CEDAR entries and the relevant publications.

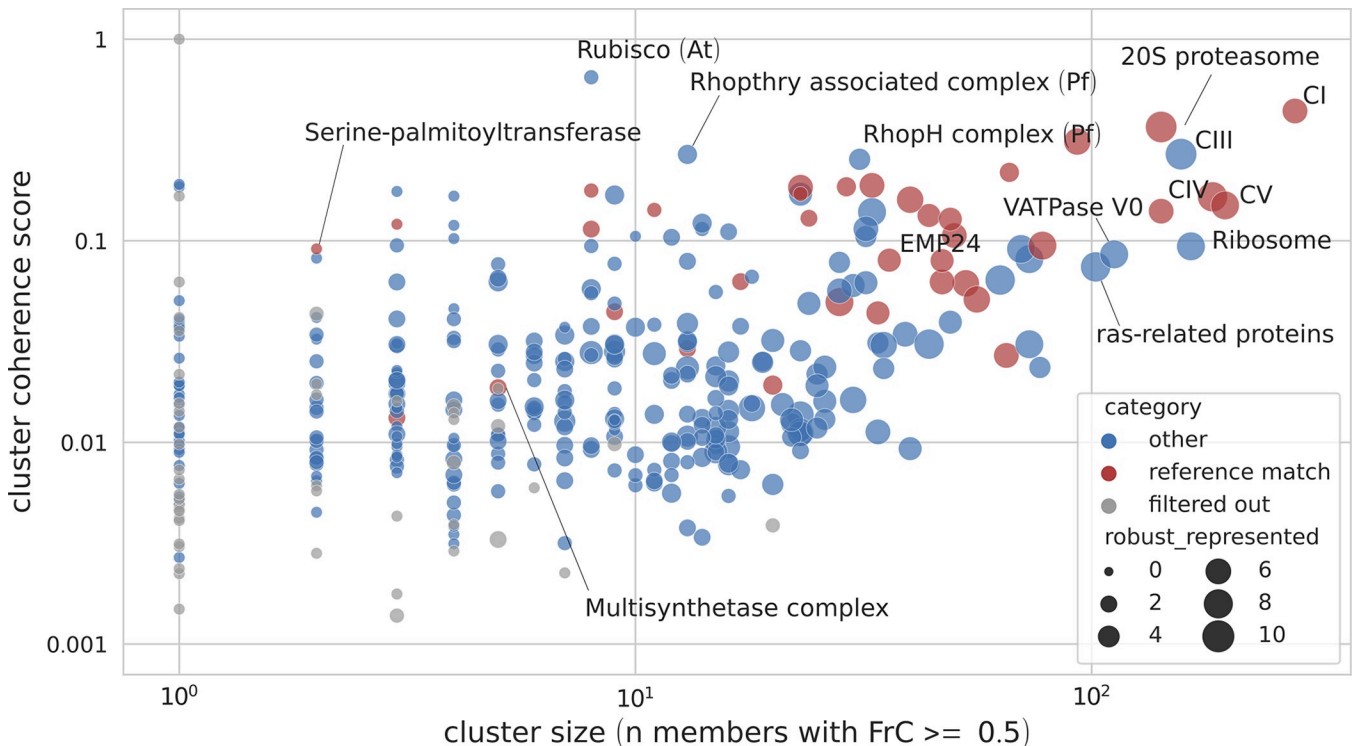

**Fig 2. Overview of clusters resulting from CompaCt analysis of a collection of complexome profiling datasets.** Each dot corresponds to an MCL supercluster, with dot size corresponding to the number of subclusters (e.g., species) consistently represented in this cluster. The x-axis represents the number of proteins consistently represented in the cluster. The cluster coherence score displayed on the y-axis is a measure for the degree of commonality between the clustered proteins per dataset. Various clusters that we identified to correspond to known protein complexes have been annotated in the figure as such. Note that the cluster consisting mainly of ras-related proteins is unlikely to represent a large complex, as the comigration of multiple members of this protein family is most likely caused by their similar mass.

complexome, each collection of datasets representing one of the other complexomes is added to the human data, computing the MMR resulting from its inclusion. After adding the complexome data that increased the MMR the most, this process was repeated until finally data from all 12 complexomes were included. At each step the complexome data that increased the MMR the most was added to demonstrate that even after adding the most informative data in the early steps, the MMR increased further as the remaining datasets were included. In most cases the inclusion of datasets from other species resulted in a higher MMR, reflecting improved agreement of the human subclusters with the CORUM reference complexes as compared to separate analysis (Fig 3). This highlights the utility of including a diverse set of complexome data as opposed to separate analysis even if the focus is on a specific system.

Notably, while generally the recovery of human complexes improves as additional data are included in the analysis, the effect varies per complexome and across inclusion stages. Given the heterogeneity of the analysed data (e.g., resolution, number of replicates, number of detected proteins, sample processing, evolutionary distance from human etc.), we are unable to determine from this which dataset features contribute the most to improved recovery of complexes. With regards to varying benefits depending on the stage at which data is included, it is possible that lower-resolution datasets (such as those from *A. thaliana*) might initially be detrimental to the formation of well-defined modules in the hypernetwork, while they might add some value in the presence of sufficient high-resolution data to form clearly defined modules.

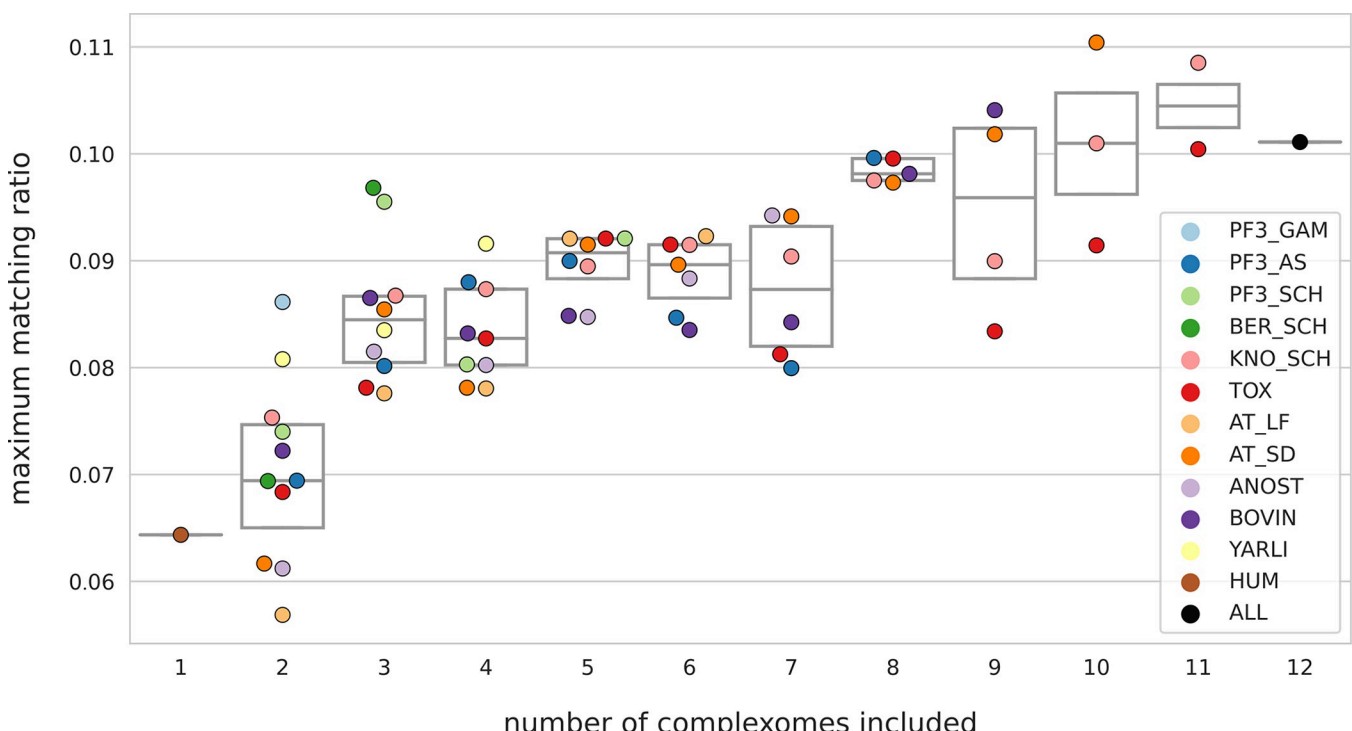

**Fig 3. Effect of including data from additional complexomes to CompaCt analysis on the agreement of *H. sapiens* cluster results with CORUM v4 [17].** Starting with only datasets representing the human complexome, datasets from additional complexomes were added to the analysis one by one, computing the MMR after including each. The remaining datasets are then added in a progressive manner, prioritizing those that resulted in the highest MMR increase. The y-axis displays the maximum matching ratio (MMR) between the human subclusters and the CORUM reference set of protein complexes, reflecting the overlap of the identified clusters with the reference. At each step the MMR resulting from including data from each remaining complexome is shown.

To our knowledge CompaCt is unique in its ability to perform combined clustering of PPI data from multiple complexomes, while allowing the composition of clusters to vary per complexome, thus enabling identification of taxon-specific elements. However, to compare the performance of CompaCt with existing approaches that aim to identify complexes from a single complexome, we compare it to the performance of ClusterONE (27), a state-of-the-art method commonly used to identify protein complexes from protein interaction data [7–9,28]. We applied ClusterONE to the human protein interaction datasets used as input for CompaCt. Fig A in S1 Appendix shows the agreement of the ClusterOne output clusters with CORUM using optimized parameters, compared to the CompaCt results (Fig A in S1 Appendix, Supplemental methods in S1 Appendix). CompaCt performance is comparable to ClusterONE when applied to the human complexome data, but it outperforms ClusterONE when applied to the complete set of complexomes.

## CompaCt allows comparative analysis of complexes through robust identification of conserved as well as taxon-specific subunits

To demonstrate that CompaCt is able to accurately recover the composition of complexes from the analysis of complexome profiling data, including taxon-specific subunits, we focused on the five well studied and resolved oxidative phosphorylation (OXPHOS) complexes in *H. sapiens*, *Y. lipolytica* and *A. thaliana*. The ability to include multiple datasets representing the same complexome allows CompaCt to assign membership confidence scores to cluster members using the FrC score. Therefore, we determined the recovery of the OXPHOS complexes

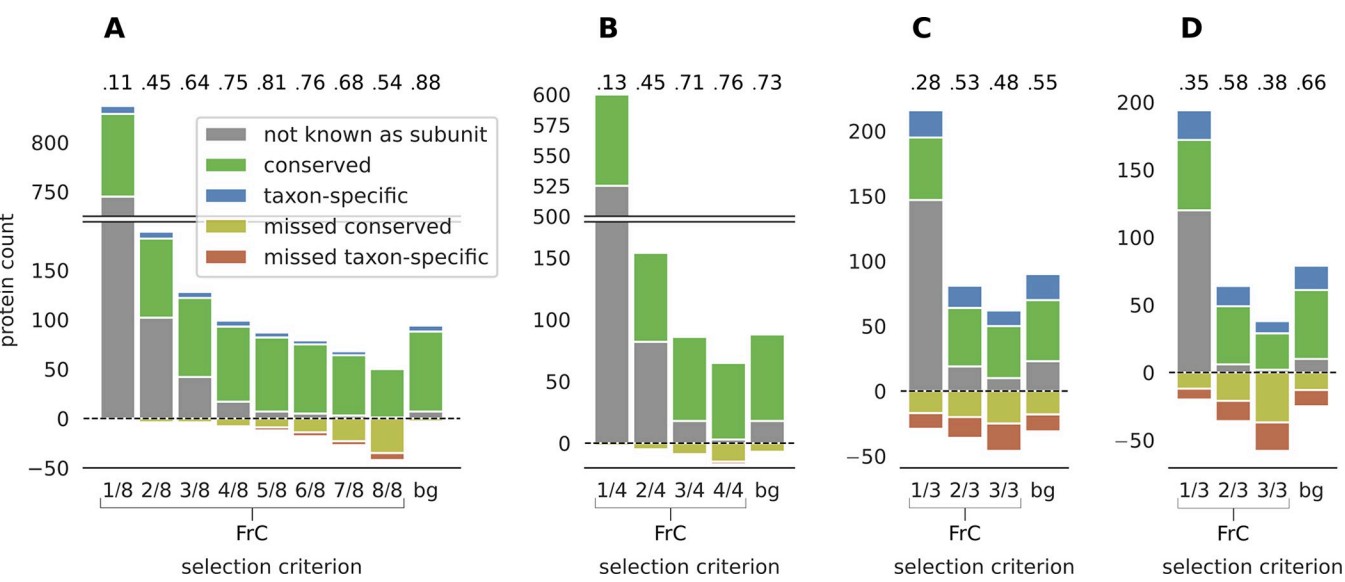

**Fig 4. Recovery of the five oxidative phosphorylation complexes by the corresponding subclusters in four complexomes.** The x axis categories correspond to different protein inclusion criteria to the subclusters. The fractions reflect the minimum fraction clustered (FrC) score for proteins to be included: i.e., in how many datasets per complexome this protein is part of the cluster. The "bg" ("best guess") selection criterion includes all proteins that have a fraction clustered of over 1/2. In addition to those, it also includes proteins with lower FrC scores, but that cluster with an orthologous protein scoring higher than 1/2. Bars above zero correspond to counts of all proteins included in the corresponding clusters. Bars below zero correspond to known complex members that are present in at least one dataset but are not part of the corresponding cluster (false negatives). The numeric values above the bars represent Jaccard index values (true positives divided by true positives, false positives and false negatives) for each selection criterion. (A) *H. sapiens*, (B) *Y. lipolytica*, (C) *A. thaliana* seedling, (D) *A. thaliana* leaf.

by their corresponding clusters while varying the FrC-based inclusion criteria of cluster members (Fig 4). These results show that selecting proteins based on a mimimum FrC greatly reduces inclusion of false positives, while losing a limited number of known subunits. To quantify this, we computed the ratio of recovery of true positives over false positive and missed scores (i.e., the Jaccard index) for each selection criterion. Aside from providing the FrC scores for cluster members, CompaCt implements an additional criterion to include likely relevant cluster members, termed the "best guess" selection: in addition to proteins that themselves meet the FrC threshold, other clustered proteins are also included when they have an ortholog or equivalent in one of the subclusters corresponding to another complexome that does meet the FrC threshold. The recovery of OXPHOS complexes using this criterion results in the highest Jaccard index in all but one complexome (Fig 4). Thus, the ability to prioritize proteins based on consistent comigration in datasets from its own species or one of the other species is helpful in discerning true complex members from spuriously clustered proteins.

The recovery of complex I, the largest of the OXPHOS complexes, was analyzed in detail to illustrate that CompaCt's output can be used to reconstruct the composition of a complex in multiple species (Fig 5). Out of a total of 149 known subunits in these three species, 135 were detected in at least one of the analyzed datasets. Of these, 128 were part of the best guess selection of the corresponding cluster, four were part of the cluster but fell below the threshold, and three subunits were not part of the Complex I cluster. 11 proteins are part of the best guess selection of the cluster while not being a known subunit (HS:0 YL:6, AT:5). The two Arabidopsis proteins that do not cluster with the other complex I subunits, At1g65290 and At3g08610, were both only detected in only one out of six Arabidopsis datasets, and in these data show visibly different migration than other complex I subunits. *Y. lipolytica* sulfur transferase (ST1) that has been observed in cryo-EM structures to be associated with complex I [29] does not

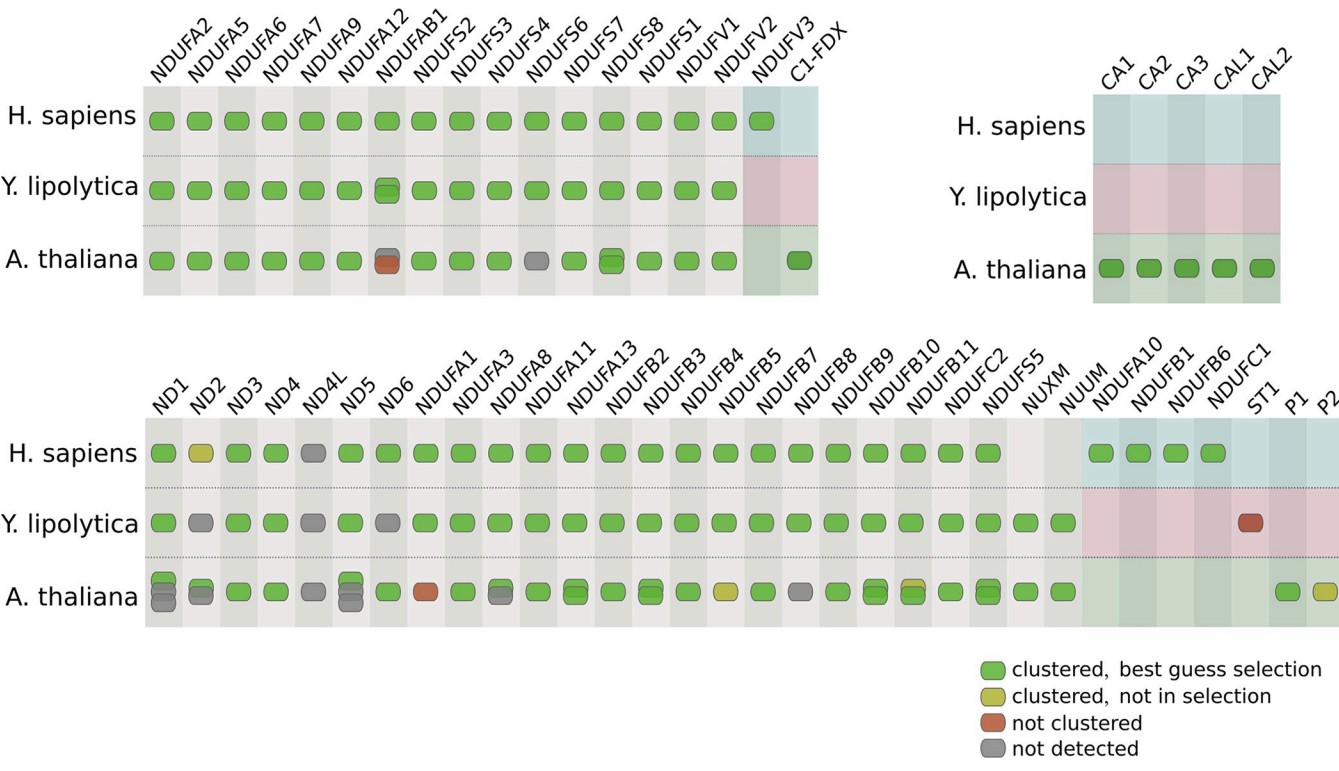

**Fig 5. Recovery of the respiratory chain complex I in *H. sapiens*, *Y. lipolytica* and *A. thaliana* by the corresponding CompaCt output cluster.** Subunits in green are detected and are part of the cluster. Subunits in yellow were not detected in any of the analyzed complexome profiles. Subunits in red were detected in at least one complexome profiling dataset, but are not part of the cluster. Taxon-specific elements of the protein complex are depicted against a colored background, the others correspond to conserved elements. Proteins in columns are orthologs or best hit homologs.

cluster with the other complex I subunits. Since ST1 was found only in a subset of particles and genetic deletion of its gene does not affect complex I assembly and function [30] it is considered to be a sub-stoichiometric and non-essential subunit of *Y. lipolytica* complex I. Like the two *A. thaliana* proteins, its migration in the complexome profiling datasets differs from the other subunits. This is not surprising since ST1 is known to dissociate from complex I during BN-PAGE [31]. Aside from these three, we were able to recover all other 132 detected complex I subunits across these three species, including 13 taxon-specific subunits that are only present in one of the analyzed species.

## Large-scale analysis of complexome profiles pinpoints novel candidate complexes and interactors

All datasets used in this analysis, except those from *A. stephensi*, have been studied separately before, often to answer a research question regarding a specific complex or set of complexes. Despite the fact that most analyzed complexome profiles are generated from samples enriched for mitochondria, they also consistently capture many complexes located in other compartments of the cell. One of the aims of this study is to leverage the systematic approach and increased sensitivity from integrative analysis of these data with CompaCt to identify novel candidate interactors and protein complexes. To this end, we inspected a selection of 45 clusters resulting from the analysis that likely represent (partial) protein complexes (Table 2). The "best guess" selection members of these clusters, from seven species whose complexomes were previously studied, comprise a total of 1603 proteins. By evaluation of the UniProt records of

**Table 2. Overview of 45 inspected superclusters.**

| Species | Clusters | Assoc. with complex | Not assoc. with complex | Not assoc., subunit ortholog | Total proteins |
|---|---|---|---|---|---|
| *A. thaliana* | 23 | 136 | 83 | 2 | 221 |
| *B. taurus* | 24 | 169 | 21 | 0 | 190 |
| *H. sapiens* | 38 | 340 | 37 | 3 | 380 |
| *P. berghei* | 21 | 143 | 15 | 3 | 161 |
| *P. falciparum* | 28 | 209 | 39 | 3 | 251 |
| *P. knowlesi* | 24 | 165 | 14 | 5 | 184 |
| *Y. lipolytica* | 27 | 181 | 34 | 1 | 216 |
| All species total | 45 | 1343 | 243 | 17 | 1603 |

Overview of 45 inspected superclusters resulting from CompaCt analysis of eukaryote complexome profiling data, filtered using the "best guess" cluster member selection criterion. The second column shows how many of the 45 inspected clusters contain proteins from each analyzed species. Protein members are divided in three categories based on information available in their Uniprot [32] entries: proteins that have evidence associating them with the respective protein complex, proteins that have no evidence associating them with the complex, and proteins that have no evidence associating them with the complex, but we predicted based on sequence-based homology to be orthologous with a known subunit of the respective complex.

these proteins we determined that for 1343 of these proteins there is already evidence associating them with the corresponding protein complexes, ranging from sequence-based predictions to experimental evidence [32]. The remaining 260 entries in UniProt contain no evidence of involvement with the respective complexes. Of these, 17 clustered proteins were identified as orthologs of known complex subunits in one of the other analyzed species, suggesting that their function has been conserved. A detailed overview of the identified members per cluster and species is available in S2 Data. We will discuss three of the analyzed clusters in more detail: the emp24 complex, the $F_1F_o$-ATP synthase complex and the vacuolar type ATPase complex.

## Evidence for an evolutionary conserved p24 complex

Members of the p24 protein family have been found to play a role as cargo receptors in anterograde as well as retrograde vesicular ER-Golgi trafficking. In yeast it has been shown that members of this protein family form a heteromeric protein complex, consisting of emp24p, erv25p, and most likely erp1p and erp2p [33,34]. These four proteins consistently clustered together in *Y. lipolytica*, supporting the hypothesis that they form a heteromeric complex (Fig 6A). The supercluster containing the *Y. lipolytica* p24 proteins also contains *H. sapiens* and *P. falciparum* subclusters with members of the p24 protein family (Fig 6A). Phylogenetic analysis of all the members of this protein family (InterPro ID: IPR015720)[35] in these three species shows that all but one (TMED6) of the human representatives of this family were detected in the analyzed complexome profiles, and are part of the same cluster. While interactions between some of the human p24 proteins have previously been shown and are e.g., present in the protein-protein interaction database STRING [36], the proteins have not yet been assigned to a multisubunit complex in *H. sapiens*. The apparent mass of the detected protein complex estimated from the human complexome profiles is ~110–120 kDa (Fig B in S1 Appendix), approximately matching the mass of a putative heteromeric complex of the five clustered subunits (125 kDa). TMED4, TMED5 and TMED7 result from recent gene duplications at the root of the vertebrates (Fig 6B). They are part of the p24 cluster, but do not make the "best guess" selection threshold because they do not cluster as consistently with the complex and are not predicted as one-to-one orthologs with one of the other clustered proteins. Nevertheless, their presence in the cluster and their homology suggests that they may sub-stoichiometrically replace their respective in-paralogs in a subset of the human p24 complexes.

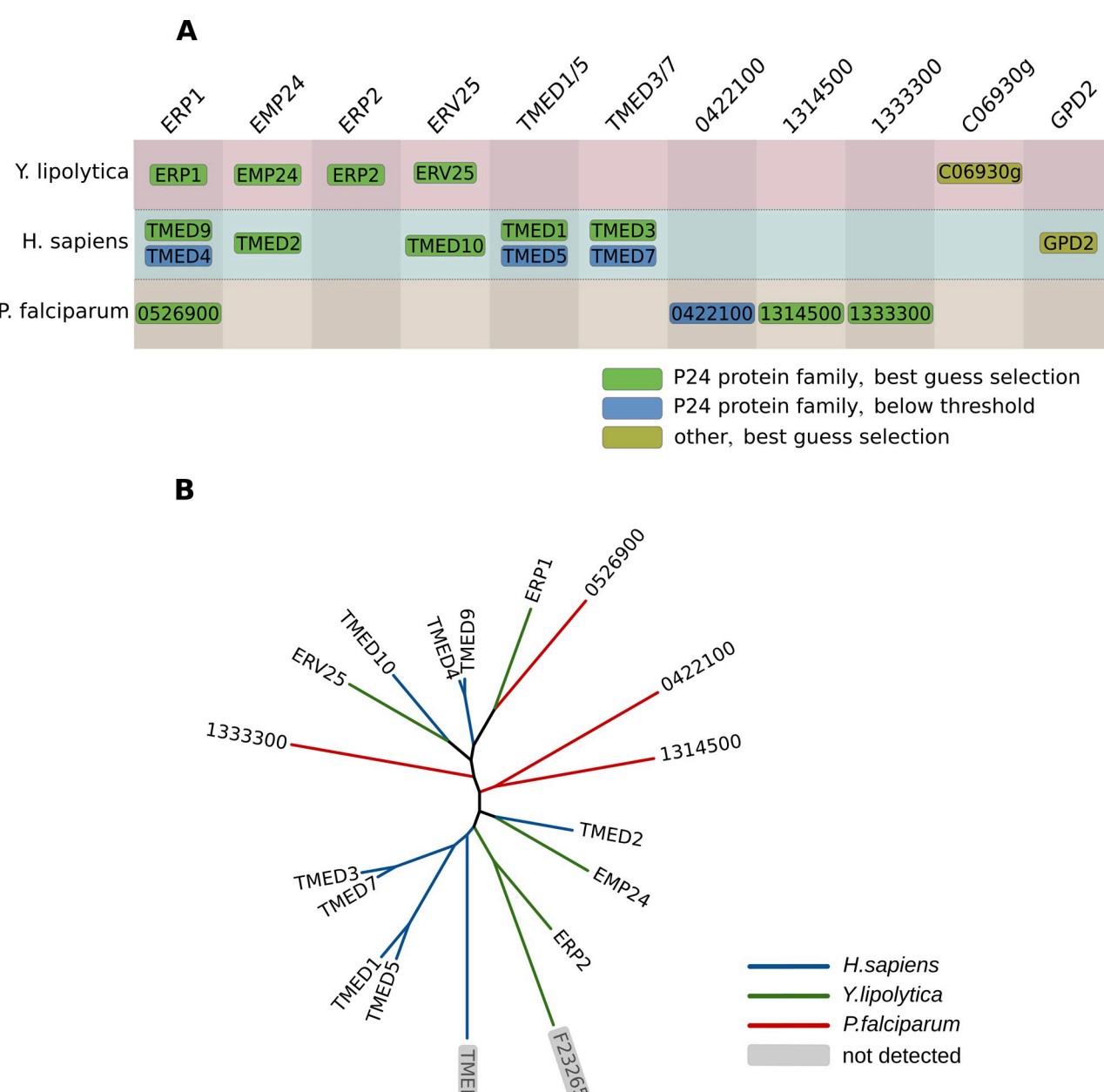

**Fig 6. p24 protein cluster overview.** (A) Overview of CompaCt output cluster containing proteins of the p24 family in *Y. lipolytica*, *H. sapiens* and *P. falciparum*. Clustered proteins with a fraction clustered score of over 0.5 or best hit homologs of those are displayed. Proteins in columns are orthologs or best hit homologs with two or more proteins in the same block, reflecting likely gene duplications. (B) Phylogenetic tree of all representatives of the p24 protein family in the three aforementioned species. Species-specific branches have been colored. Members of the protein family that were not detected in the data used in this project are shown in gray. Sequences were aligned with ClustalOmega [39], and the phylogenetic tree was reconstructed with PhyML [40]. The dynamic evolution of the protein family whose members are part of one complex are well captured by the CompaCt approach.

The *P. falciparum* subcluster contains all four *P. falciparum* representatives of the p24 protein family (Fig 6A), but no specific investigation of the p24 complex in malaria parasites has been performed. Two of these (PF3D7_0526900, PF3D7_0422100) have however been shown to interact in a pairwise assay and to be essential for liver-stage viability or sporozoite infectivity [37]. Notably, PF3D7_13333300 has been predicted to interact with an apicoplast-resident

protease, which seems inconsistent with a role in ER-Golgi protein trafficking [38]. To what degree the *Plasmodium* emp24 complex function is conserved and whether there is a specific function in the liver or sporozoite stages remains to be explored.

## High confidence assignment of ATP synthase subunits

The $F_1F_o$-ATP synthase complex is a highly conserved complex central to energy conversion in the mitochondria of eukaryotes and bacteria and has been well studied in all model species. In multiple complexomes, many members of this complex form a supercluster (Fig 7). Two human cluster members were not reported as ATP synthase subunits previously, but one of them (C15orf61) was recently observed to comigrate with ATP synthase subunits by Morgenstern *et al.* in a complexome profiling experiment [41]. We could confirm that C15orf61 consistently co-migrates with ATP synthase across multiple replicates and studies, as it is the only non-ATP synthase subunit cluster member with an FrC as high as 6/8. While it has been annotated as a secreted protein based on experimental data, it is predicted to be mitochondrial protein in Mitocarta 3.0 based on MS/MS and APEX labelling experiments and on the presence of a predicted mitochondrial target sequence [42]. To determine whether absence of C15orf61 affects functioning of the ATP synthase complex, we generated a C15orf61 knock-out in human HEK cells. Enzymatic activities of the five individual OXPHOS complexes were not significantly affected by ablating C15orf61, although the activity of complex I tended to be somewhat lower (Fig C in S1 Appendix). IGF2BP2 (FrC: 5/8) is not predicted to be a mitochondrial protein, and has been ascribed a function in mRNA binding and transport [43]. The *P. knowlesi* ATP synthase subcluster contains four proteins that have not been described previously as subunits of this complex. Three of those subunits: PKNH_0117300, PKNH_0725400 and PKNH_1124100 are orthologous to known *T. gondii* ATP synthase subunits, suggesting that their function as complex V subunits is conserved among apicomplexa. Supporting this, PKNH_0725400 and PKNH_1124100 are predicted to be mitochondrially located in PlasmoMitoCarta [44]. Lastly, the uncharacterized TGGT1_233890 protein, one of the *T. gondii* cluster members not previously described as a subunit, has been assigned to the inner mitochondrial membrane [45], consistent with our assignment.

## The V-ATPase complex outside the Metazoa

Vacuolar type ATPase (V-ATPase) is a highly conserved eukaryotic proton pump that is responsible for acidification of a variety of intracellular compartments or whole cells. The

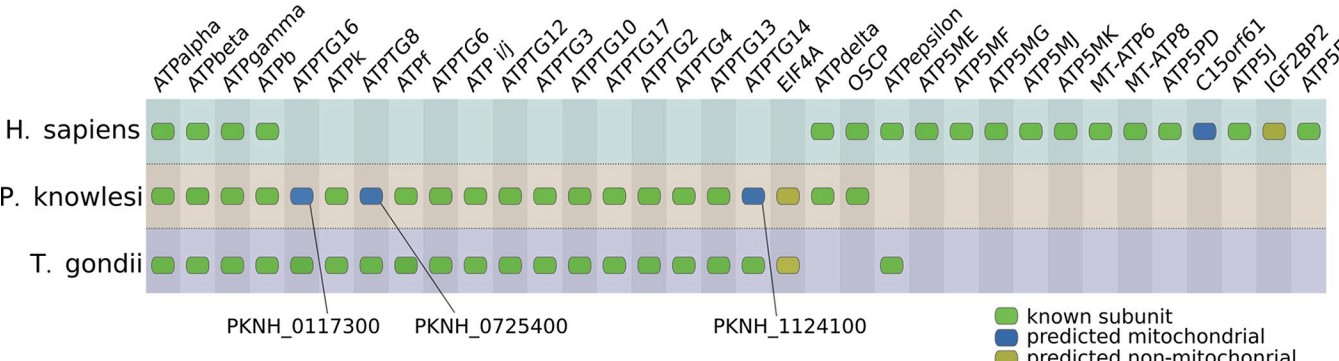

**Fig 7. Overview of CompaCt output cluster containing subunits from the mitochondrial ATP Synthase complex in *H. sapiens*, *P. knowlesi* and *T. gondii* complexomes.** Clustered *P. falciparum* and *H. sapiens* proteins with a fraction clustered score of over 0.5 or best hit homologs of those are displayed, as well as their clustered *Toxoplasma* orthologs. Human proteins labeled "predicted mitochondrial" are listed in Mitocarta 3.0 [42]. Plasmodium and Toxoplasma proteins with this label are predicted as such by PlasmoMitoCarta [44] and a HyperLOPIT study [45], respectively.

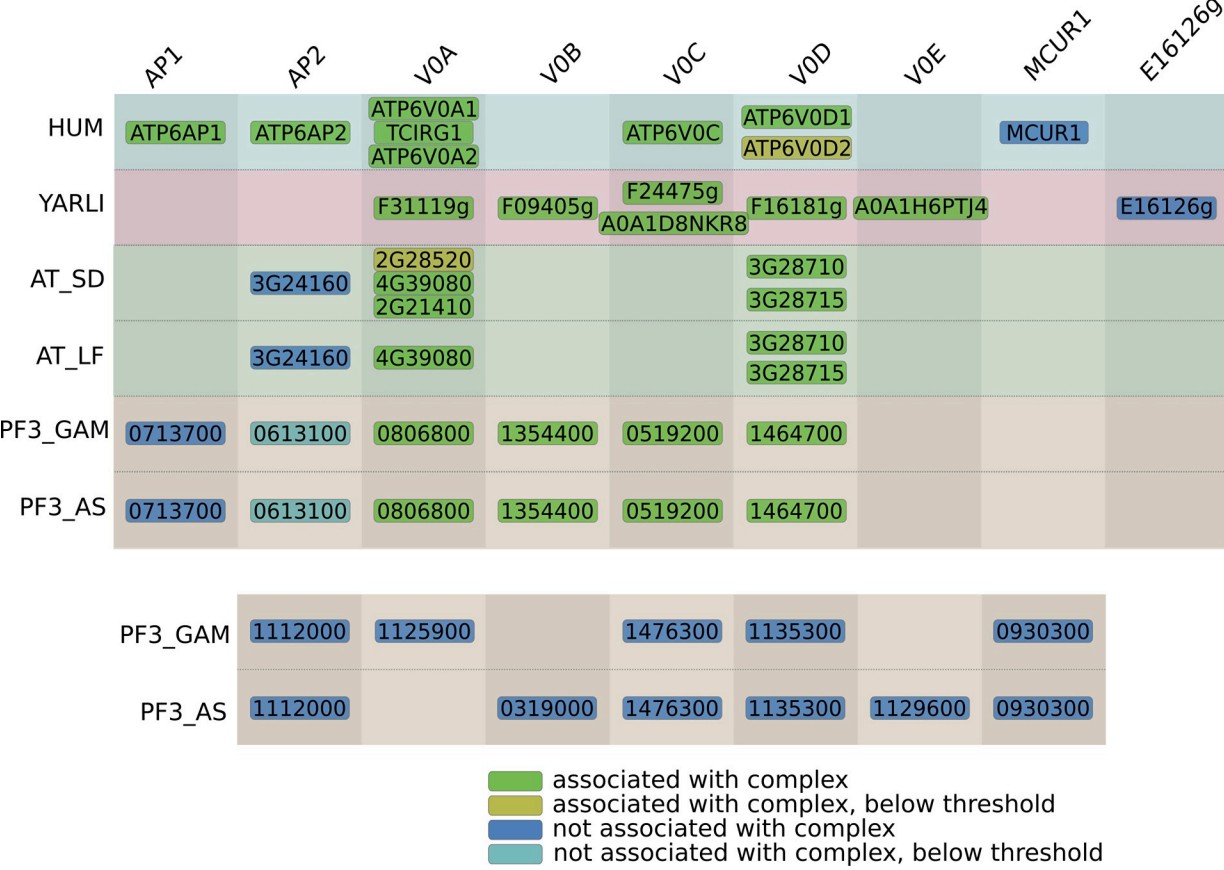

**Fig 8. Overview of CompaCt output cluster containing subunits from the membrane component ($V_0$) of the Vacuolar ATPase complex (V-ATPase) in *H. sapiens*, *Y. lipolytica*, *A. thaliana* and *P. falciparum* complexomes.** Clustered proteins with a fraction clustered score of over 0.5 or best hit homologs with those are displayed. Proteins in columns are orthologs or best hit homologs, and are labeled depending on whether they have previously been associated with this protein complex. Two or more proteins in the same block indicate likely gene duplications. For proteins labeled as "associated with the complex" there is some previous evidence associating them with the V-ATPase complex, while for the other proteins there is not.

complex is subdivided into two modules, the membrane bound $V_0$ module and the hydrophilic $V_1$ module. The $V_0$ module of this complex is represented in a separate supercluster (cluster 7) from the $V_1$ module supercluster (cluster 13, S1 Data), caused by the differing migration patterns of the two modules, which in turn is likely due to dissociation during the experimental procedure. Of the $V_0$ module the *H. sapiens*, *Y. lipolytica*, *A. thaliana* and *P. falciparum* subclusters are shown in Fig 8. We present the first characterization of the *P. falciparum* V-ATPase complex based on experimental data. While some of the detected subunit identities are clear from sequence-based homology, our cluster contains several potential novel subunits of this complex that are either taxon-specific or whose orthology could not be established through standard homology detection methods.

Although two members of this cluster were annotated as proteins of unknown function, profile-based homology searches identified them as orthologs of the human V-ATPase subunit ATP6AP2: at3g24160 (*A. thaliana*) and PF3D7_0613100 (*P. falciparum*). Orthologs of ATP6AP2 were not identified previously as subunits of V-ATPase outside of the Metazoa. Another protein that is part of the *P. falciparum* subcluster is PF3D7_0713700, which we identify as a likely ortholog of human V-ATPase subunit ATP6AP1, even though it was not

predicted to have a human ortholog as the hhsearch result was above the threshold (E-value = 1.6). It should be noted that ATP6AP1 evolves at a very high rate, as also the sequence identity between the human and the *Saccharomyces cerevisiae* ATP6AP1 proteins is very low [46]. Additionally, we predict A0A182XX75 as the *A. stephensi* ortholog of ATP6AP1, which is also part of the cluster corresponding to the $V_0$ component of V-ATPase. Notably, the C-terminal transmembrane (TM) region is the only part that is conserved in this protein family, whose lack of sequence conservation has been established before [46,47]. We now extend ATP6AP1 to *Anopheles* and *Plasmodium*, of which the *Plasmodium* ATP6AP1 is only 209 amino acids long with two predicted TM helices (positions: 5–25, 168–190), compared to 239 amino acids in human, with a single TM region after cleavage of the original protein at the furin cleavage site [47]. An alignment of the C-terminal TM region of ATP6AP1 and its orthologs in a number of species in shown in Fig D in S1 Appendix. The *Y. lipolytica* subcluster contains YALI0_F09405g, a protein that is predicted to contain a "V-ATPase proteolipid subunit" domain (IPR035921) in the InterPro database [35], but whose complex membership had not yet been established. Additionally, several proteins that have no orthologs in one of the other analyzed species are part of this cluster. The human MCUR1 protein has a known function as a regulator of the mitochondrial calcium uniporter and is localized to mitochondria [48], so it is not likely to be part of this complex. The *Y. lipolytica* protein YALI0_E16126g has no known function, but is co-expressed with known subunits as indicated in the STRING database [36] and thus a potential interactor of the *Y. lipolytica* V-ATPase complex. Finally, several V-ATPase candidates are included in one of the two *P. falciparum* complexomes, of which only PF3D7_1112000 has a high fraction clustered score (6/8) in both. PF3D7_1112000 is a small (72 amino acid) protein of unknown function that is predicted to contain two transmembrane helices that take up the majority of the protein, suggesting a possible role in the $V_0$ module of the V-ATPase complex.

## CompaCt analysis of novel complexome sheds light on respiratory chain complex evolution

To demonstrate the suitability of CompaCt for the characterization of an unstudied complexome, we generated and included a dataset published with this study from the mosquito *A. stephensi*. This is the first large-scale analysis of complexome profiling data from a species from the Protostomes, one of the two major taxa of the Bilateria. We provide experimental evidence for the origin of the constituents of multiprotein complexes that hereto were mainly based on their orthology to experimentally characterized proteins from the Deuterostomes, the other major taxon that contains the vertebrates. We generated two datasets as byproducts from isolation of *P. falciparum* sporozoites from *A. stephensi* salivary glands and detected 925 proteins of the mosquito host of the parasite. Table A in S1 Appendix provides an overview of putative *A. stephensi* complex subunits identified through analysis with CompaCt. From the superclusters corresponding to known complexes in one of the well-studied species like *H. sapiens* and *A. thaliana*, we identified fifteen clusters that contain *A. stephensi* proteins homologous to known cluster members in one of the other species. Across these clusters we provide experimental support for 87 proteins whose association with the respective complexes were hereto solely based on sequence similarity. Additionally, we identify five proteins of unknown function that both cluster with the complex and are orthologous to a known subunit in one of the other analyzed species. The complete list of identified proteins per cluster is available in S2 Data.

One of the identified clusters corresponds to the mitochondrial respiratory chain complex cytochrome *c* oxidase (complex IV). This cluster contains four proteins predicted to be associated with this protein complex based on sequence similarity: A0A182XZQ2, A0A182YLP4,

A0A182YLP7 and A0A182Y5T1. Additionally, this cluster contains *Anopheles* protein A0A182Y1F7, the ortholog of the *Drosophila CG7630* protein, recently identified as an ortholog of mammalian cytochrome *c* oxidase subunit COX7B, which was previously believed to be vertebrate specific [49]. The alignment of COX7B, the *Anopheles* A0A182Y1F7 protein and the *Drosophila CG7630* protein is shown in Fig E in S1 Appendix. The *A. stephensi* ATP synthase cluster contains two proteins (A0A182YCB1, A0A182YSV3) that are best hit homologs with the human beta subunit, and two proteins (A0A182Y5P7, A0A182YCU3) that are best hit homologs with the epsilon subunit, indicating that these two genes have been duplicated and that all four have retained their role as subunits of complex V.

## Discussion

The increasing availability of complexome profiling data from various species in the public CEDAR database has opened the door for large scale and automated analysis of these data [13]. However, methods that enable automated large-scale comparative analysis of protein interaction data to exploit this wealth of information were lacking. In this work, we demonstrate the potential of large-scale integration of protein interaction data from multiple species with CompaCt. Our approach of integrating interaction data has several benefits over existing methods. Firstly, comparing datasets using interactor profiles avoids direct comparison of migration patterns, circumventing difficulties caused by batch effects introduced through differences in protocols and complex separation resolution hampering direct comparisons. Secondly, indirect usage of orthology to determine similarity between two proteins' set of interactors rather than directly connecting orthologous proteins enables incorporation of proteins with no known orthologs and therewith the detection of taxon-specific interaction members or the proteins that have diverged beyond what can be detected by sequence-based homology detection methods. An example of the former is the emp24 complex supercluster, where we determined the putative composition of this complex in *H. sapiens*, *P. falciparum* and *Y. lipolytica*, even though it varies between these species and contains taxon-specific elements. This demonstrates that the dynamic evolution of the protein family whose members are part of one complex are well captured by our approach. An additional benefit of this approach is that it exhibits robustness to incomplete orthologies or imperfect correspondence of protein identifiers between datasets. We found that in many such cases CompaCt is still able to recover these proteins and interactions based on orthology between the interactors. For example, a number of mitochondrially encoded complex I subunits were correctly clustered, even though their orthology was missed because of mismatched identifiers between the various complexome data and the proteomes used for orthology predictions (Fig E in S1 Appendix).

One of the main challenges in the identification of complexes from complexome profiling data is the occurrence of spuriously comigrating proteins or protein complexes, as thousands of identified proteins are separated into typically just 60 fractions. We demonstrate that by systematically prioritizing cluster members based on consistency of interactions, enabled through inclusion of multiple interaction datasets from the same species, we were able to greatly simplify identification of novel potential interactors that were missed in previous individual analyses of the datasets used. For example, we identified a number of novel candidate interactors in the V-ATPase complex, of which PF3D7_0613100 and AT3G24160 are predicted as orthologs of human V-ATPase subunit ATP6AP2 through profile-based homology (Fig 8). Similarly, PF3D7_0208800 and PF3D7_0505900 consistently cluster with the ER membrane and the prefoldin complexes respectively, and are predicted as orthologs of human subunits of these complexes (EMC10, PFDN6) using profile-based homology (S1 Data).

Stacey et al. have recently shown that clustering of protein-protein interaction networks is susceptible to noise [50]. They found that part of the resulting clusters are stable and robust to noise while others are not, with the former more often biologically relevant, and propose a perturbation strategy to determine the stability of clusters. Rather than using a perturbation approach to identify likely biologically relevant and stable clusters, CompaCt leverages the fact that it combines multiple datasets to determine the consistency with which proteins and their orthologs from different datasets are clustered together, captured by the "cluster coherence" score.

CompaCt integrates data from multiple complexomes in a single network and then clusters them together, after which complexome-specific clusters are extracted. We show that this has two major advantages over separate analyses in which proteins are assigned to complexes based on data from a single complexome. Firstly, conserved protein complexes present in multiple species will generally be grouped together as subclusters of the same cross-species supercluster, foregoing the need to a posteriori match or align clusters from separate analyses. More importantly, we demonstrate that our method of combined clustering of multiple complexomes results in overall better recovery of protein complex composition, by showing the agreement with a known reference improves when including additional complexomes in the analysis. E.g., protein complexes that are only partially detected in a dataset will often be difficult to detect, but can still contain novel interactors missed in other experiments. An example of this are the three candidate *Plasmodium* ATP synthase subunits identified in this study, that were missed in the initial publications despite being orthologous to known *T. gondii* ATP synthase subunits. Evers et al. analyzed the mitochondrial complexome of gametocyte and asexual blood-stage parasites, where they described the composition of ATP synthase but failed to detect these proteins [24]. Hillier et al performed large-scale clustering analysis of whole-cell schizont complexome profiles, where they did not identify a well-defined ATP synthase cluster, as the complex was well represented in only one of their replicates [9]. In our combined analysis, ATP synthase was represented in multiple complexomes, resulting in a well-defined ATP synthase cluster, which included the three subunits detected in the data from Hillier et al, that remained undetected in the complexome profiles of Evers et al.

By inherently allowing transfer of information between species in an unsupervised manner, CompaCt simplifies large-scale characterization of previously unstudied complexomes. This is demonstrated by the identification of a number of complexes and interactors in the *A. stephensi* complexome. Automated large-scale characterization of complexes in previously unstudied species can shed light on complex evolution. While the evolutionary origin of subunits can be inferred based solely on the distribution of orthologs [51], this approach has two pitfalls. Firstly, it may not be able to detect the homolog through sequence-based methods. An example of this is A0A182XX58, an *A. stephensi* protein that clusters with complex I; its closest human homolog is the complex I subunit NDUFA3, but it was not identified as orthologous in our analysis because the E-value resulting from HHsearch did not meet the significance threshold (hhsearch E-value = 0.056). While drafting this manuscript, the *D. melanogaster* ortholog of this protein was identified as an ortholog of NDUFA3 based on Cryo-EM data [52], supporting this finding. Secondly, the presence of an ortholog in a species does not necessarily indicate that it is part of the same complex. As an example, the *P. falciparum* protein ApiCOX13 (PF3D7_1022900) has recently been identified as a subunit of complex IV, and is part of our complex IV cluster. However, using sequence-based homology this protein can be identified as orthologous to human protein CISD3, which is not part of human complex IV. The *Anopheles* subcluster corresponding to respiratory chain complex I contains A0A182YAZ9, the *A. stephensi* ortholog of the mammalian complex I subunit NDUFA10. While this protein has orthologs throughout the Metazoa, it is unclear when this protein

became part of complex I, as evidence suggests a nucleoside kinase function outside of its role as complex I subunit and the protein binds dGTP [53]. Notably, this protein is homologous to deoxyribonucleoside kinases, and its active sites have been conserved [54]. Furthermore, its migration in human and bovine complexome profiles (Fig F in S1 Appendix) suggest that it is, besides being part of complex I, also present as a monomer, pointing towards a potential dual role as monomeric enzyme and subunit of complex I. Our results indicate that this protein is part of complex I in Protostomes, suggesting it was already a subunit of complex I at the root of the Bilateria. The migration of this protein in *A. stephensi* does not clearly indicate it is present as a monomer (Fig F in S1 Appendix).

In this study, to score interaction likelihood between protein pairs within complexome profiles we used the Pearson correlation. This metric is most commonly used and generally performs very well to determine interaction partners from complexome profiles. However, recent work has suggested and demonstrated the effectiveness of several other metrics or machine learning-based scores to determine interactions [7,8,11,28,55]. Here we refrained from using those, as our focus was on the method of integrating interaction datasets rather than identifying the best within-dataset interaction metric. Additionally, several of these metrics rely on external evidence and a reference set of known complexes, which are not available for some of the less well-studied species. However, as these other metrics might be better suited in certain situations to determine interaction likelihood, it is important to note that the CompaCt software is able to analyse interaction matrices previously scored with any similarity metric.

In conclusion, the approach of comparing protein interaction data between species presented here enables more complete comparative analyses of multiple complexomes than existing methods. While CompaCt was used to analyze multiprotein complexes and their interactors here, it could also be used for the comparative analysis of other types of interaction data like gene co-expression data or to identify modules of coregulated genes.

## Methods

### CompaCt software

**Input data.** CompaCt performs integrative cluster analysis on multiple protein interaction datasets representing one or more biological systems (e.g., species, tissue, cell-type etc.).

An interaction dataset consists of a set of elements that represent proteins, with real number scores between each pair of elements representing interaction likelihood, which can be represented as a symmetric matrix of correlation or interaction scores. Any numeric metric or score that reflects interaction likelihood or strength and allows ranking of each protein's interactors can be used. Optionally, the raw element expression/abundance data (e.g.: protein abundances per fraction in case of complexome profiles) can be provided, in which case CompaCt automatically computes Pearson's correlation scores between element pairs. The provided datasets are subdivided into "collections", where each collection should represent a distinct biological system in which we expect the protein interactions to be identical to each other (e.g., human fibroblast, *Arabidopsis* leaf, *Plasmodium* mitochondria, etc.). Inclusion of multiple datasets representing the same collection can be included to more robustly identify true interactors, by prioritizing consistently clustering proteins. Multiple datasets representing the same species are expected to use the same protein identifiers, but are allowed to contain different sets of elements of varying sizes. In order to be able to compare and identify common elements between datasets from different species, pairwise, "one-to-one" orthologies between these species need to be provided. A pairwise orthology between species is represented as a set of identifier pairs, one from each species, that represent orthologous proteins. It is not required that every protein present in the data of one of the species has an ortholog in any of the other species.

**Comparing interactor profiles with rank biased overlap.** For each element in each dataset, a ranked list of all other elements within this dataset, ordered from highest interaction score to lowest, is determined. This ranked list is referred to as the molecule's interactor profile. To be able to determine for two proteins from different interaction datasets to which degree they have a common set of interacting proteins, we determine the similarity between their interactor profiles, using the non-extrapolated rank biased overlap (RBO) metric [14] (github.com/ changyaochen/rbo). RBO is a so-called set-based overlap metric, that assigns a value between 0 and 1 representing the degree of similarity between the ranking of two lists of elements.

The RBO metric has two properties that make it suitable for the comparison of interactor profiles. First, the ranked lists can contain elements not common in both lists and are allowed to have different lengths. This is required for the comparison of interactor profiles, as often a different set of proteins is identified in different datasets, and when comparing between species it is likely that part of the proteins will not have an ortholog in the other species. Secondly, RBO is a "top-heavy" score, assigning more weight to the top of the list when determining overlap, with a tunable parameter that determines the extent to which the metric is "top-heavy". A focus on the high-scoring interactors makes biological sense, as only the top scoring proteins are likely to be true interactors of the given protein. This RBO top-heaviness parameter is a tunable parameter in the CompaCt tool. To determine a suitable default value for this parameter for the analysis of complexome profiling data, we determined the effect of changing this parameter on recovering pairs of proteins belonging to the same protein complex (Fig G in S1 Appendix), showing that a parameter value of 0.9 results in the highest median rank of all protein pairs part of the same complex, and that the rank biased overlap is able to prioritize protein pairs that are part of the same complex over other protein pairs.

To limit computational cost, the search depth when computing RBO scores, i.e., the length of the ranked lists, is limited so that lower ranks that have very little influence on the RBO score are not considered. By default, the contribution of each rank is computed, and the ranks that cumulatively contribute 99% of the RBO score are used. To compute rank biased overlap between interactor profiles from different species, pairwise orthologies are used to determine the common elements between the ranked lists. Proteins from different species that are orthologous are considered common elements when computing RBO scores, while the others are considered unique to that list.

## Reciprocal top hits

Pairwise all-to-all comparisons between 2 datasets result in a large number of RBO scores, of which the majority will not represent proteins with actual shared interactors. To reduce the amount of data and CPU time that is considered in the next steps of the CompaCt workflow we want to exclude uninformative similarity scores. However, absolute RBO score values representing a true set of shared interactors vary greatly, because proteins from a small protein complex will result in much lower RBO scores than proteins that are part of a large complex, due to the difference in the number of shared interactors. Therefore, to select relevant RBO scores when comparing two datasets we determine whether for both proteins the RBO score between them is in the top 1% (adjustable parameter in CompaCt) of all scores for the protein in question. If this is the case, we refer to it as a "reciprocal top hit", and the edge between this pair is included in the hypernetwork used for cluster analysis.

## Combined network clustering

The approach described above is used to systematically determine interactor profile similarity of all proteins between two datasets. This is then performed in a pairwise manner comparing

each dataset to all others regardless of which collection (i.e., a number of datasets representing the same species/biological system) they are a part of. To perform combined clustering of all datasets, similarity scores originating from all pairwise dataset comparisons should be comparable. However, as different datasets likely have a different number of common proteins, the similarity scores will likely vary between the pairs of datasets compared. Therefore, before these scores are combined and then clustered, they are normalized such that they have the same mean. After computation of similarity scores, selection of reciprocal top hits and normalization, the resulting hypernetwork connecting proteins from all included datasets is clustered using the MCL cluster algorithm version 14–137 [15] using the default parameters.

## Processing clusters: prioritizing proteins

The resulting clusters, extracted from the hypernetwork by MCL, can contain elements originating from multiple datasets and collections. These super-clusters are first split into subclusters separating elements by input collection. The subclusters, containing elements from datasets belonging to the same collection, are then further processed to assign a single "fraction clustered" (FrC) score to each unique protein that represents the consistency with which it is part of a cluster. Per protein and subcluster, the number of elements from different datasets representing this protein are counted, and divided by the total number of datasets in the corresponding collection (e.g.; if 4 datasets are provided for a given collection, and the current subcluster contains elements corresponding to this protein for 3 out of 4 of those datasets, the fraction clustered score for this protein in this cluster equals ¾). Note that to compute the FrC, the total number of datasets from that complexome is used as the denominator instead of the number of datasets that this protein is detected in, as we consider a protein not being detected lack of evidence of interaction.

## Processing clusters: prioritizing clusters

Additionally, to determine the coherence score of a supercluster, the fraction of possible matches is computed, reflecting the degree to which the same or orthologous proteins from different datasets are clustered together. A match is defined as: two "equivalent" elements from different datasets are part of the same supercluster. Equivalent elements are either two elements representing the same protein in two datasets from the same species, or two orthologous proteins from different species. The total number of matches found within a supercluster is then divided by the total number of possible matches given the cluster's composition, to get the fraction of possible matches. Consider that a cluster contains n proteins from dataset A, and k proteins from dataset B. The number of possible matches would then be the equal to the minimum value of n and k. In the case where a cluster contains proteins from more than two datasets, the possible number of matches is the sum of possible matches between each dataset pair. In addition to computing a coherence score for each complete supercluster, CompaCt computes the fraction of possible matches for each specific subcluster, to reflect the coherence of each cluster in that specific system. As an example, the computation of the fraction of possible matches for the CompaCt supercluster containing the rubisco complex is illustrated in Table 3. The composition of this supercomplex is shown in Table B in S1 Appendix. To calculate the fraction of possible matches the total number of actual matches (46) is divided by the total number of possible matches (71), resulting in a score of 0.648. To remove clusters that are not likely to be biologically relevant, CompaCt performs a filtering step, retaining only clusters with a minimum of 2 matches as well as having at least one protein with a fraction clustered of at least 0.5.

**Table 3. Rubisco supercluster (cluster id: 410) actual and possible matches.**

| Comparison | | Actual matches | Possible matches |
|---|---|---|---|
| CRS100 | CRS101 | 2 | 4 |
| | CRS102 | 2 | 3 |
| | CRS103 | 2 | 4 |
| | CRS104 | 2 | 4 |
| | CRS105 | 2 | 4 |
| CRS101 | CRS102 | 3 | 3 |
| | CRS103 | 3 | 7 |
| | CRS104 | 5 | 7 |
| | CRS105 | 3 | 6 |
| CRS102 | CRS103 | 3 | 3 |
| | CRS104 | 3 | 3 |
| | CRS105 | 3 | 3 |
| CRS103 | CRS104 | 5 | 8 |
| | CRS105 | 4 | 6 |
| CRS104 | CRS105 | 4 | 6 |
| | **total** | 46 | 71 |

## Annotation of clusters with reference complexes

To simplify identification of protein complexes from the output clusters, CompaCt implements automatic annotation of clusters using a provided reference. For one of the input collections a collection of reference complexes can be provided. For each provided reference complex, the cluster that contains the most members of this complex is identified. If a large enough fraction of the complex is present in the cluster, for which a threshold can be provided, it will be annotated with the name of the reference complex.

## Analysis of eukaryote complexomes

**Complexome profiling.** Salivary glands containing *P. falciparum* sporozoites were collected from female *A. stephensi* mosquitoes (Sind-Kasur Nijmegen strain) and consequently homogenized [56,57]. The homogenate was then either separated into gland material and *P. falciparum* sporozoite material or kept unseparated. The resulting protein samples were solubilized and subsequently separated using BN-PAGE [58]. A complete description of this process is available in the Supplemental methods in S1 Appendix. In-gel trypsin digestion followed by mass spectrometric analysis was performed as described in [24].

The remaining complexome profiles used in this paper are reused from published studies and have been made publicly available on the CEDAR database, from which they were retrieved. A detailed overview of the used datasets including CEDAR accessions is available from Table C in S1 Appendix. The protein abundances were used as is, no normalization or preprocessing was performed before analysis with CompaCt. Identifiers were converted to match those used in orthology predictions.

**Cell culture conditions.** HEK293T cells (ATTCC, 293T-CRL-3216) were cultured in High glucose DMEM medium supplemented with 10% fetal calf serum, 1% Penicillin/Streptomycin and 1% Sodium pyruvate. Cells were cultured at 37˚C and 5% CO2.

## Generation of KO HEK293T cells

C15orf61 was knocked-out in HEK293T using CRISPR/Cas9 system, followed by clonal expansion and characterization. The characteristics of the resulting clones used in this work is

**Table 4. Characteristics of the different c15orf61 HEK293T clones used in this work.**

| Clone | Alias | Passage | Allele 1 | Allele 2 | gRNA used |
|---|---|---|---|---|---|
| HEK293T not transfected | FM-WT HEK | | WT | WT | None |
| Clone 2E7 | FM-2-E7 | 29 | WT | WT | gRNA1 |
| Clone 4E7 | FM-4-E7 | 29 | WT | WT | gRNA1+2 |
| Clone 2D4 | FM-2-D4 | 29 | c.-8_132delins83 p.Met1? | c.-101_95delins45 p.Met1? | gRNA1 |
| Clone 2E11 | FM-2-E11 | 29 | c.-149_103delins149 p.Met1? | c.-149_103delins149 p.Met1? | gRNA1 |
| Clone 3C10 | FM-3-C10 | 29 | c.26_*243del p.(Ala11Cysfs*21) | c.26_*243del p.(Ala11Cysfs*21) | gRNA1+2 |
| Clone 4F5 | FM-4-F5 | 29 | c.-152_*267del p.Met1? | c.-152_*267del p.Met1? | gRNA1+2 |

shown in Table 4. A complete description of the methods is available in the Supplemental methods in S1 Appendix.

## Enzyme activity measurements

Enzyme activity measurements of respiratory chain complexes in clonal cell-lines were performed as previously described [59]. Presented activity values are relative to the activity of respiratory chain complex II.

## Orthology prediction

Pairwise orthologies can be provided as input to CompaCt, alongside interaction datasets, to allow comparison between interaction data from different species. To sensitively determine pairwise orthology between all species included in this project, we determined the bidirectional best hits between their respective proteomes. The proteomes used in the homology searches are listed in Table D in S1 Appendix. To determine bidirectional best hits between two proteomes, protein blast searches were performed in both "directions". Any proteins that did not have a significant (Evalue < 0.001) hit, were subsequently searched using hhsearch from the hhsuite toolkit using default parameters [60]. The set of best hits from blast was complemented with any significant best hit hhsearch results. This combined set of best hits was used to determine bidirectional best hits between each pair of proteomes. The blast software version 2.9.0 was used. Custom blast databases were generated from the aforementioned proteomes. All-to-all blast searches were run against these databases using blastp with default settings. The hhsuite software version 3.0 was used. Custom profile databases were generated from the proteomes used in each species, following the protocol available from (https://github.com/soedinglab/hh-suite/wiki#building-customized-databases), skipping the secondary structure annotation as recommended. Profile based homology searches were run with hhsearch using default settings.

## Annotation and evaluation of results using CORUM reference

For automatic annotation of CompaCt's output clusters, a reference set of known protein complexes can be provided. For this we used the complete human complexes from CORUM v4 [17]. To evaluate the agreement of CompaCt's resulting clusters with a set of human reference complexes, we performed some further processing on the human CORUM reference. Firstly, any duplicate complexes have been removed, so that each complex occurs once. Additionally, any complexes that are a subset of other complexes, generally corresponding to sub-assemblies, were removed. To enable fair assessment of the clustering approach, rather than of the completeness of the underlying data, any proteins not detected in at least two of the eight human datasets were removed from the reference. Lastly, any protein complexes containing

two or fewer proteins (after the previous steps) were removed from the reference, to focus on multiprotein complexes that are well represented in the underlying data. To quantitatively assess the agreement of CompaCt's output clusters with the aforementioned reference, we used the maximum matching ratio, as described in [27].

### Analysis of eukaryote complexome profiles with CompaCt

The complexome profiles described earlier were analyzed together with the CompaCt command line tool, using default settings. The aforementioned pairwise orthologies were included to allow comparison between species, and the CORUM reference was included for automatic annotation of the resulting clusters. In order to assess the effect of including data from additional complexomes on the human cluster results, several additional CompaCt analyses were run using various combinations of complexome datasets. First, a single collection of eight human complexome profiling datasets corresponding to the human complexome was analyzed. Then, in separate runs, the human datasets were analyzed together with each of the remaining complexomes. The combination that resulted in the best agreement of human subclusters with the CORUM reference was selected for further analysis. This process was then repeated progressively, until all complexomes were included in the final analysis.

### Annotation of cluster members in inspected clusters

To determine whether protein members of the analyzed result clusters have previously been associated with the respective protein complexes, their UniProt entries were inspected for any evidence associating them with the protein complexes represented by the respective clusters (last accessed on Nov 3, 2022) [32]. Any type of evidence suggesting involvement with this protein complex was deemed sufficient, be it experimental evidence or automated predictions. Additionally, some proteins that lacked evidence of involvement on UniProt but for which we were aware of recent evidence showing association with the corresponding complex were categorized as such.

## Supporting information

**S1 Appendix. Supplemental Figures A-G, supplemental tables A-H, supplemental methods.**
(PDF)

**S1 Data. Output clusters from CompaCt analysis of eukaryote complexome profiles.**
(XLSX)

**S2 Data. Annotation of inspected clusters and cluster members.**
(XLSX)

## Acknowledgments

We would like to thank Marga van de Vegte-Bolmer, Rianne Stoter and Wiebe Kooijman for generating the *P. falciparum* parasites and the mosquito infections. We would like to acknowledge Astrid Pouwelsen, Saskia Mulder, Jolanda Klaassen, Laura Pelser-Posthumus and Jacqueline Kuhnen for breeding of the mosquitoes and for all the salivary gland dissections to obtain the sporozoites.

## Author Contributions

**Conceptualization:** Joeri van Strien, Martijn A. Huynen.

**Data curation:** Joeri van Strien, Felix Evers.

**Formal analysis:** Joeri van Strien.

**Funding acquisition:** Ulrich Brandt, Martijn A. Huynen.

**Investigation:** Felix Evers, Madhurya Lutikurti, Stijn L. Berendsen, Alejandro Garanto, Geert-Jan van Gemert, Alfredo Cabrera-Orefice, Richard J. Rodenburg.

**Methodology:** Joeri van Strien.

**Software:** Joeri van Strien.

**Supervision:** Alejandro Garanto, Taco W. A. Kooij, Martijn A. Huynen.

**Visualization:** Joeri van Strien.

**Writing – original draft:** Joeri van Strien, Felix Evers, Martijn A. Huynen.

**Writing – review & editing:** Joeri van Strien, Felix Evers, Alejandro Garanto, Alfredo Cabrera-Orefice, Richard J. Rodenburg, Ulrich Brandt, Taco W. A. Kooij, Martijn A. Huynen.

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
