## [Decision Letter · Decision Letter 0]

28 May 2023

Dear Dr. Huynen,

Thank you very much for submitting your manuscript "Comparative Clustering (CompaCt) of eukaryote complexomes identifies novel interactions and sheds light on protein complex evolution" for consideration at PLOS Computational Biology. As with all papers reviewed by the journal, your manuscript was reviewed by members of the editorial board and by several independent reviewers. The reviewers appreciated the attention to an important topic. Based on the reviews, we are likely to accept this manuscript for publication, providing that you modify the manuscript according to the review recommendations.

Sincerely,

Dina Schneidman

Academic Editor

PLOS Computational Biology

Nir Ben-Tal

Section Editor

PLOS Computational Biology

Reviewer's Responses to Questions

**Comments to the Authors:**

Reviewer #1: The manuscript presents a new tool for deriving protein complex predictions from CF-MS experiments. I really like the approach of using orthology between species to try to build better interactomes. I have a few questions/concerns:

1. What exactly is the input data? E.g., will CompaCt take output from PrInCe? I get that it takes pairs of proteins with some kind of likelihood score but more information is needed there.

2. The Foster group has shown that clustering interactome data to derive complex predictions is extremely susceptible to noise (https://pubmed.ncbi.nlm.nih.gov/33592499/). How does CompaCt get around this issue? It appears that only a single type of clustering is applied, with no way to control for noise introducing errors

3. How does CompaCt perform compared to other algorithms? I was expecting to see some side-by-side analysis demonstrating that this gives some advantage

Reviewer #2: The manuscript by Joeri van Strien, Felix Evers, Madhurya Lutikurti, Stijn L. Berendsen, Alejandro Garant, Geert-Jan van Gemert, Alfredo Cabrera-Orefice, Richard J. Rodenburg, Ulrich Brandt, Taco W.A. Kooij and Martijn A. Huynen describes a method named CompaCt that allows the comparison of complexome profiles across various experiments and different species. The authors describe very well the methodology of native protein preparation and separation before describing the methodological challenges to comparatively analyze multiple complexome profiles, that differ in preparation, separation and detection methods. By converting similarities between protein profiles within data sets in ranked lists of decreasing local similarity, CompaCt can compare profile similarities across various experiments. After filtering and normalization of the RBO score matrix, a network is generated and clustered by MCL. Metrics to interpret the relevance of specific proteins or clusters are provided as fraction_clustered and cluster coherence respectively. By including various data sets of different tissues/species, CompaCt can distinguish between consistently co-migrating proteins and spuriously co-migrating proteins, and provides interesting insights into new putative evolutionary conserved complexes.

The method presented by the authors is very interesting and intriguing and the manuscript is well written. However, there are some questions that remain unclear after reading the manuscript in its current form.

Comments:

1. In “Comparing interactor profiles with rank biased overlap“ it is stated that “the rbo metric is a set-based overlap metric, that assigns a degree of overlap between two ranked lists”. Sets do not inherit any particular element order. It should be clarified, that the ranking is a key property for this rank similarity metric.

2. In Processing clusters: prioritizing clusters it is stated that “The total number of matches found within a supercluster is then divided by the number of possible matches given the cluster’s composition, to get the fraction of possible matches.” It is ambiguous how the total number of matches is determined. Is it the number of possible edges, or is the orthologue distribution taken into account? How is a coherence of 0.65 obtained for rubisco given the supplemental data. While the point size and x-axis location match the table, the coherence score is not clear.

3. Please elaborate, why both Arabidopsis samples in the first 3 inclusion steps are the most detrimental for the MMR, but afterwards perform as top candidates in recovering CORUM complexes.

Minor:

o In SF2_selected_clusters.xlsx the sheet “45_selected_clusters”, Column “fraction_present”

Float decimal points are corrupted

**Have the authors made all data and (if applicable) computational code underlying the findings in their manuscript fully available?**

Reviewer #1: Yes

Reviewer #2: None

PLOS authors have the option to publish the peer review history of their article (what does this mean?). If published, this will include your full peer review and any attached files.

Reviewer #1: **Yes: **Leonard Foster

Reviewer #2: No

Figure Files:

Data Requirements:

Reproducibility:

References:

---

## [Decision Letter · Decision Letter 1]

10 Jul 2023

Dear Dr. Huynen,

We are pleased to inform you that your manuscript 'Comparative Clustering (CompaCt) of eukaryote complexomes identifies novel interactions and sheds light on protein complex evolution' has been provisionally accepted for publication in PLOS Computational Biology.

Best regards,

Dina Schneidman

Academic Editor

PLOS Computational Biology

Nir Ben-Tal

Section Editor

PLOS Computational Biology

Reviewer's Responses to Questions

**Comments to the Authors:**

Reviewer #1: The authors have addressed my concerns

Reviewer #2: As a reviewer, I have carefully examined the manuscript and the revisions made by the authors in response to my comments. I would like to recommend accepting the manuscript for publication based on the thoroughness and effectiveness with which the authors have addressed my suggestions.

Throughout the revision process, the authors have taken into account each comment raised and have made appropriate modifications to the content, and presentation of the research. They have addressed both major and minor concerns, ensuring that the manuscript now provides a more comprehensive and clearer presentation and higher level of detail regarding the method description.

Based on the authors' diligent and comprehensive response to my comments, I strongly recommend accepting the manuscript for publication. I am confident that this work will make a valuable contribution to the existing body of knowledge in the field and will stimulate further discussion and research.

**Have the authors made all data and (if applicable) computational code underlying the findings in their manuscript fully available?**

Reviewer #1: Yes

Reviewer #2: None

PLOS authors have the option to publish the peer review history of their article (what does this mean?). If published, this will include your full peer review and any attached files.

Reviewer #1: **Yes: **Leonard Foster

Reviewer #2: No

---

## [Editor Report · Acceptance letter]

31 Jul 2023

PCOMPBIOL-D-23-00550R1 

Comparative Clustering (CompaCt) of eukaryote complexomes identifies novel interactions and sheds light on protein complex evolution

Dear Dr Huynen,

I am pleased to inform you that your manuscript has been formally accepted for publication in PLOS Computational Biology. Your manuscript is now with our production department and you will be notified of the publication date in due course.

With kind regards,

Timea Kemeri-Szekernyes
